# Epigallocatechin-3-Gallate Improves Intestinal Gut Microbiota Homeostasis and Ameliorates *Clostridioides difficile* Infection

**DOI:** 10.3390/nu14183756

**Published:** 2022-09-11

**Authors:** Zhengjie Wu, Jian Shen, Qiaomai Xu, Qiangqiang Xiang, Yunbo Chen, Longxian Lv, Beiwen Zheng, Qiangqiang Wang, Shuting Wang, Lanjuan Li

**Affiliations:** 1State Key Laboratory for Diagnosis and Treatment of Infectious Diseases, National Clinical Research Centre for Infectious Diseases, Collaborative Innovation Centre for Diagnosis and Treatment of Infectious Diseases, The First Affiliated Hospital, Zhejiang University School of Medicine, 79 Qingchun Rd., Hangzhou 310003, China; 2Jinan Microecological Biomedicine Shandong Laboratory, Jinan 250021, China

**Keywords:** *Clostridioides difficile*, EGCG, metabolites, gut microbiota, transcriptome

## Abstract

*Clostridioides difficile* infection is closely related to the intestinal flora disorders induced by antibiotics, and changes in the intestinal flora may cause the occurrence and development of *Clostridioides difficile* infection. Epigallocatechin-3-gallate (EGCG) is one of the major bioactive ingredients of green tea and has been suggested to alleviate the growth of *C. difficile* in vitro. EGCG can ameliorate several diseases, such as obesity, by regulating the gut microbiota. However, whether EGCG can attenuate *C. difficile* infection by improving the gut microbiota is unknown. After establishing a mouse model of *C. difficile* infection, mice were administered EGCG (25 or 50 mg/kg/day) or PBS intragastrically for 2 weeks to assess the benefits of EGCG. Colonic pathology, inflammation, the intestinal barrier, gut microbiota composition, metabolomics, and the transcriptome were evaluated in the different groups. Compared with those of the mice in the CDI group, EGCG improved survival rates after infection, improved inflammatory markers, and restored the damage to the intestinal barrier. Furthermore, EGCG could improve the intestinal microbial community caused by *C. difficile* infection, such as by reducing the relative abundance of *Enterococcaceae* and *Enterobacteriaceae*. Moreover, EGCG can increase short-chain fatty acids, improve amino acid metabolism, and downregulate pathways related to intestinal inflammation. EGCG alters the microbiota and alleviates *C. difficile* infection, which provides new insights into potential therapies.

## 1. Introduction

*Clostridioides difficile* infection (CDI) is a common result of antibiotics-induced diarrhea and may lead to death if left untreated. Health care costs and mortality increase as a result of frequent recurrence, as well as reduced quality of life [1,2]. When broad-spectrum antibiotics are administered and the commensal microbiota in the intestine is depleted, pathogens such as *C. difficile* can exploit this vulnerability [3,4]. *C. difficile* infection is associated with dysbiosis of the gut microbiota [5].

Epigallocatechin-3-gallate (EGCG) is an anti-inflammatory, antioxidant, antitumor, cancer chemopreventive, and antiaging factor [6,7,8,9]. Catechins in green tea are believed to be a polyphenolic flavonoid with biological activity [10], and 50 and 80% of the catechin content comes from EGCG. EGCG has received increasing attention for its antioxidant and health-promoting effects [11,12,13]. In vitro experiments have shown the protective effect of EGCG against *C. difficile* [14]. A growing number of evidence shows that this EGCG alleviated DSS-induced murine colitis [15], obesity [16], and radiation-induced intestinal injury [17] by improving the intestinal microbiota. Gut microbes are closely related to diet in terms of composition and function [18]. The intestinal microbiota has effects in the multifaceted physiological functions of the host. The active connection between the gut microbiota, antioxidative, and inflammatory pathways is important to the host’s health. However, whether the gut flora plays a role in EGCG-mediated *C. difficile* infection remains unclear.

Our study examined the survival of infected mice after EGCG administration, evaluated the protective effect of EGCG against intestinal infection, and examined the underlying mechanism of EGCG-mediated intestinal protection. We found that EGCG increased survival, reduced weight loss, improved intestinal pathological damage and barrier damage, restored the gut microbiota, and changed the metabolism in mice. In addition, we found that the protective mechanism may include inflammatory signaling pathways. In conclusion, our findings suggest the potential use of EGCG in clinical therapies, for example as a dietary supplement prebiotic during infection.

## 2. Materials and Methods

### 2.1. Animals and Treatments

C57BL/6 female mice aged 6 to 8 weeks were randomly grouped: (1) control group (CON group, n = 8) was administered PBS orally once daily, (2) *C. difficile* infection group (CDI group, n = 12), (3) oral administration of 25 mg/kg body weight EGCG (EL group, n = 10), and (4) 50 mg/kg EGCG (EH group, n = 10) from day -8 to day 5 daily. The EGCG (Sigma-Aldrich, St Louis, MO, USA) doses were based on the literature [15]. For *C. difficile* infection (CDI group, EL group, and EH group), the mice were administered a cocktail of antibiotics for 5 days [19] (Figure 1A), which included colistin (850 U/mL), gentamicin (0.035 mg/mL), kanamycin (0.4 mg/mL), metronidazole (0.215 mg/mL), and vancomycin (0.045 mg/mL), followed by to the administration of plain sterile water for 2 days. On day -1, the animals were administered clindamycin (10 mg/kg, ip). Then, 10^8^
*C. difficile* strain VPI 10463 (ATCC 43255) was gavaged into the mice on day 0. Feces samples were collected before the mice were sacrificed.

### 2.2. Histology, Immunohistochemistry, and Immunofluorescence Analysis

Proximal colon tissue was collected after the mice were killed and was fixed in formalin for paraffin embedding. Three-micrometer sections were stained with hematoxylin and eosin for morphological examination. Histological scores were assessed by epithelial cell damage (0–3 points), congestion/edema (0–3 points), and neutrophil margination (0–3 points) [19]. Immunohistochemistry (Ly6G and F4/80) and immunofluorescence analysis of ZO-1 was performed according to a previous protocol [20].

### 2.3. Hematological Examination of Inflammatory Markers

Cytokines in the mouse serum were detected using the Bio-Plex Pro Mouse Cytokine 23-Plex Panel (Bio-Rad). Mouse LBP ELISA kits (Abcam, Cambridge, UK) were used to measure the LPS binding protein (LBP) concentrations.

### 2.4. RT–PCR

Colon tissue was extracted with the RNeasy Plus Mini Kit to extract RNA (Qiagen, Valencia, CA, USA). qPCR analysis was performed using Premix Ex Taq (Takara Biomedicals, Kusatsu, Japan) and a ViiA7 real-time PCR system (Applied Biosystems, Waltham, Massachusetts, USA). β-actin acted as an internal control. A list of primer sequences can be found in Appendix A.

### 2.5. 16S rRNA Sequencing

Fecal DNA was collected from the mice prior to execution and was isolated using the DNA PowerSoil Kit (Qiagen, Valencia, CA, USA). The V3-V4 variable region was amplified using universal primers (Appendix A). The PCR products were extracted, purified, and sequenced on Illumina NovaSeq platform. More details of the sequencing protocol are provided in the Appendix A. Raw sequencing data were filtering, denoised, merged, and detected using DADA2 [21] with QIIME2 [22]. Alpha and beta diversity analyses were performed using QIIME2 software. Sequencing data for the 16S rRNA gene are available in NCBI’s Sequence Read Archive (SRA) database (PRJNA867379).

### 2.6. Metabolic Profiling

As previously described [23], analysis of the cecum contents was performed using gas chromatography mass spectrometry (GC-MS). The samples were centrifuged and filtered after being dissolved in a cold methanol solution, and then vacuum freeze-dried, methoxylated, and mixed with heptadecanoic acid. A GC-MS system from Agilent 7890A-5975C was used for the detection of the samples. The NIST database was used to identify metabolites. Orthogonal partial least squares discriminant analysis (OPLS-DA) was performed with SIMCA software to identify metabolic differences.

The concentration of short-chain fatty acids (SCFAs) in the cecum contents was quantified by GC–MS [24]. Briefly, the samples were homogenized with ultrapure water and were centrifuged. The filtrate was mixed and incubated with ethyl acetate. Then, we collected the ethyl acetate layer, and the mixture was transferred to a sample vial for analysis.

### 2.7. Transcriptome Analysis

Colon tissue was extracted with a TRIzol reagent to obtain the total RNA. cDNA libraries were purified, constructed, and sequenced using an Illumina NovaSeq. After filtering using Trimmomatic, sequencing reads were mapped to the reference genome (GRCm39) by hisat2 (v2.2.1.0) [25]. The DESeq2 R package (version 1.20.0) was used to analyze the differential expression [26]. Based on Benjamini and Hochberg’s method, the *p* values were adjusted. The KEGG pathway reference database (http://www.genome.jp/kegg/, accessed on 9 March 2022) was used for the KEGG enrichment analysis. RNA sequencing raw data are available in the SRA database (PRJNA867511).

### 2.8. Statistical Analysis

The data are presented as the mean ± SEM and were analyzed using GraphPad Prism 9.0.0.0 (San Diego, CA, USA). In order to make multiple comparisons, we used one-way ANOVA and Tukey’s multiple comparison test. *p* < 0.05 was considered statistically significant.

## 3. Results

### 3.1. Oral EGCG Administration Reduces C. difficile-Induced Clinical Symptoms

EGCG was tested to determine whether it can alleviate *C. difficile*-induced colitis. Mice were gavaged with two different doses of EGCG (EL and EH group) daily (Figure 1A). As shown in Figure 1, oral EGCG significantly attenuated infection compared with that in the CDI group, which could be observed by changes in survival (Figure 1B; CDI vs. EH group, *p* < 0.05; no difference between CDI and EL group), weight loss (Figure 1C), and the alleviation of colonic shortening in mice. The improvements were more pronounced in response to high concentrations than low concentrations of EGCG.

A retrospective histological analysis of the colon tissues demonstrated that EGCG protected against intestinal injury caused by infection. The results showed that CDI group exhibited various inflammatory features, such as a thickened colonic epithelium, inflammatory cell accumulation, structural incompleteness, ruptured crypt glands, and the infiltration of inflammatory cells after infection (Figure 1D). In contrast, the structure of the colon was well protected in the EGCG-treated mice. As shown in Figure 1E, histological scores were reduced in the EGCG-treated mice compared with those in the CDI group. EGCG alleviated histological changes, such as a reduction in the edema and epithelial damage. In conclusion, EGCG ameliorated structural damage to the intestine caused by *C. difficile* infection.

### 3.2. Oral EGCG Inhibits C. difficile-Induced Intestinal Mucosal Disruption and Improves Barrier Function

When *C. difficile* spores sprout in the human gut, they produce toxins and destroy host cells [27]. Maintaining the integrity of the intestinal mucosa is essential for maintaining intestinal mucosal barrier function, which blocks the migration of intestinal microorganisms and their toxins to extraintestinal tissues and organs [28]. Intestinal permeability is associated with several tight junction proteins; therefore, we measured the mRNA levels of tight junction markers. As for the CON group, the mRNA expression levels of ZO-1 and Occludin were reduced in the CDI group (Figure 2B,C). These markers increased its expression significantly after EGCG administration. In addition, most secreted mucins make up the intestinal mucosal layer, which is a dynamic and chemically complex barrier [29]. The CDI group showed significantly lower mucin gene (MUC2) transcription than the CON mice, according to our results. EGCG significantly increased the MUC2 mRNA expression (Figure 2D). In order to evaluate how EGCG affects the colon barrier function, we assessed the expression of the intestinal barrier protein ZO-1 using immunofluorescence analysis (Figure 2A), which showed that the integrity of the intestinal mucosa was impaired after *C. difficile* infection. EGCG protected the mucosal barrier and ameliorated intestinal barrier damage, and the protective effect was more obvious in the high-dose group than the low-dose group. We also evaluated the intestinal inflammatory indices Ly6G and F4/80 through immunohistochemistry. The Ly6G and F4/80 indices represent neutrophil and macrophage infiltration, respectively (Figure 2A). The results showed that EGCG decreased the inflammatory response in the intestine. LPS is a biomarker of inflammation and intestinal permeability in hosts [30], and we examined the level of serum LBP (Figure 2E). Serum immunoreactivity to the bacterial product LPS was increased in the CDI group, and supplementation with EGCG decreased the serum LBP levels in mice.

### 3.3. EGCG Reduces the Serum Inflammation in C. difficile Infection

*C. difficile* infection can activate the NFκB pathway, causing the release of proinflammatory cytokines and chemokines [31]. To assess the ability of EGCG to ameliorate inflammation in mice, we examined the serum cytokines, and *C. difficile*-infected mice showed significantly elevated serum levels of IL-1α, IL-1β, IL-6, TNF-α, MIP-1α, and MCP-1 (Figure 3). EGCG reduced the levels of cytokines that were elevated by infection. In particular, the EH group showed a better anti-inflammatory effect than the EL group. According to these results, cytokines may play a role in EGCG-mediated amelioration of *C. difficile*-induced inflammation in the organism.

### 3.4. EGCG Regulates the Microbiota Composition in the Intestine

EGCG was examined further for its potential association among effects on the intestinal pathology, the mucosal barrier, inflammation, and the structure of the intestinal microbiota. The effect of EGCG on gut microbiota composition in *C. difficile*-infected mice was examined by 16S rRNA gene sequencing. We finally obtained raw sequencing data, which were distributed between 78,017 and 81,924. The distribution of valid tags obtained after processing was between 42,923–68,025, which was used for the next analysis. The distribution of the number of ASVs in each sample was 49–535. The rarefication curve of ASVs indicate a reasonable amount and depth of sequence (Appendix A). Compared with that in the CON group, α-diversity (Shannon, Simpson, and Chao1 indices), which indicate community richness (Figure 4A), was reduced in the CDI group, while EGCG prevented the *C. difficile* infection-induced reduction in α-diversity. β-Diversity was used to examine the structural differences in the microbial communities. Principal coordinates analysis (PCoA) based on Bray–Curtis (PERMANOVA, *p* < 0.05) suggested the CON, CDI, EL, and EH microbiota structures differ significantly (Figure 4B). We examined the taxonomic abundance and dominant differential taxa at different levels (Figure 4C, and Appendix A). The results showed that the relative abundance of *Proteobacteria*, *Enterobacteriaceae*, *Enterococcaceae*, *Escherichia-Shigella*, and *Enterococcus* was increased in the CDI group.

LEfSe analysis was used to show the characteristic bacteria in the groups (Figure 4D). Compared with the CON group, *Enterococcaceae*, *Clostridiaceae*, *Peptostreptococcaceae*, *Enterobacteriaceae*, and *Morganellaceae* were enriched in the CDI group at the genus level (Appendix A), while *Bacteroidaceae*, *Muribaculaceae*, *Prevotellaceae*, *Rikenellaceae*, *Helicobacteraceae*, *Desulfovibrionaceae*, *Acholeplasmataceae*, *Erysipelotrichaceae*, *Lactobacillaceae*, *Leuconostocaceae*, *Streptococcaceae*, *Lachnospiraceae*, *Oscillospiraceae*, *Ruminococcaceae*, and *Peptostreptococcaceae* were enriched in the CON group. Additionally, comparing the CDI group with the EH and EL groups separately showed that *Enterococcaceae*, *Enterobacteriaceae*, and *Morganellaceae* at the family level were enriched in the CDI group, while EGCG increased the levels of *Lactobacillaceae* and *Lachnospiraceae* (Figure 4D, Appendix A, and Appendix A). The intestinal flora structure of mice with *C. difficile* colitis was significantly affected by the oral administration of EGCG and that EGCG increased the relative abundance of beneficial intestinal flora, such as *Lactobacillaceae* and *Lachnospiraceae*.

### 3.5. EGCG improves Metabolic Disorders and SCFAs Levels after C. difficile Infection

Furthermore, we determined whether changes in intestinal metabolites were associated with EGCG’s benefits by analyzing the metabolite composition of the cecum content in the samples. In the PLS-DA plot, each group’s metabolome profiles were clearly separated (Figure 5A), suggesting that the metabolome compositions in these four groups differed. OPLS-DA showed that the metabolite compositions differed between groups (CDI vs. EL, CDI vs. EH, CDI vs. CON group) (Figure 5B, Appendix A). To identify the metabolic profiles separating the CDI group from the EH group, metabolites that were altered by EGCG treatment were chosen using OPLS-DA (variable importance of projection (VIP) > 1) and *p* < 0.05. The heatmap shows 145 selected metabolites, which are mainly related to the following pathways: carbohydrates, amino acids, lipids, peptides, and purines (Figure 5C and Appendix A). Significant increases or decreases were observed in the metabolites related to amino and purine pathways after *C. difficile* infection. In our study, amino acids such as L-aspartic acid were decreased after EGCG treatment, while the levels of gallic acid, L-lactic acid, ascorbic acid, galactitol, catechin, D-fructose, and linoleic acid were increased after EGCG treatment. Based on differential metabolites, the KEGG pathway enrichment analysis compared the major metabolic pathways between the CDI and EH groups (Figure 5D), including central carbon metabolism in cancer and arginine biosynthesis, which are closely associated with *C. difficile* infection.

In order to determine whether EGCG affects SCFA production, we determined SCFA concentrations in the cecal contents. *C. difficile* infection significantly reduced the levels of SCFAs such as acetic acid, propionic acid, butyric acid, and 2-methylbutyric acid (Figure 5E, Appendix A), but the oral administration of EGCG (Figure 5E) improved the levels of SCFAs, and the high concentration of EGCG significantly improved the production of acetic acid, propionic acid, and butyric acid.

### 3.6. EGCG Reshapes the Transcriptome Profile of C. difficile-Infected Mice

As the EH group showed better protection than the EL group, we selected three groups (CON, CDI, and EH groups) for transcriptomic analysis of the intestine to examine which mechanisms contribute to the protective effect of EGCG against *C. difficile*-associated colitis. Differentially expressed genes were screened using *P*adj ≤ 0.05 and |log2FoldChange| ≥ 1.0. Figure 6A shows a Venn diagram of the differentially expressed genes between the different groups. There were 571 and 846 genes that were significantly up- and downregulated when comparing the CDI and CON group, respectively. The EGCG and CDI groups were compared, and 280 and 194 genes were significantly up- and downregulated by EGCG, respectively. The differentially expressed genes between the three groups were plotted as a heatmap (Figure 6B). EGCG reduced CDI-induced transcriptional upregulation of the intestinal genes, including the genes involved in inflammation-related responses (Cxcl3, Areg, Cxcl2, and Il1r2), lipid metabolism (Lpcat4 and diacylglycerol O-acyltransferase 2), and purine metabolism (phosphodiesterase 9A). EGCG attenuated the transcriptional downregulation of intestinal genes induced by *C. difficile*, including genes associated with lipid binding (Apol9a and Apol9b) and immune regulation (Ccl22, Alox15, Cxcl9, and C1qtnf3). Further KEGG pathway analysis of the differentially expressed genes suggested that the affected pathways between the CON and CDI groups included the intestinal immune network for IgA production, the TNF signaling pathway, Th17-cell differentiation, and the IL-17 signaling pathway (Figure 6C). The differential pathways between the CDI and EGCG groups included the intestinal immune network for IgA production, Th1 and Th2 cell differentiation, Th17-cell differentiation, T-cell receptor signaling pathway, cytokine–cytokine receptor interaction, and the PPAR signaling pathway (Figure 6D).

## 4. Discussion

Having a healthy gut microbiota contributes significantly to the development and progression of *C. difficile* infection [32]. Furthermore, the intestine microbiota constitutes an important factor associated with host health and environmental factors, and microbiota metabolites are also relevant to intestinal health [33]. Therapeutic strategies to modulate the gut microbiota (e.g., FMT) are important in *C. difficile* infection [34]. Daily consumption of tea can maintain a healthy gut [35]. *C. difficile* infection shows that dysregulation of the gut microbiota is characteristic of infected mice, and we showed that EGCG regulated the structure of the intestine microbiota and its metabolites and increased the production of SCFAs, thereby promoting anti-inflammatory and antioxidant effects in the gut.

Our results showed that EGCG therapy remitted *C. difficile* infection, as indicated by a decrease in the incidence of diarrhea and an increase in survival. Previous studies have shown that EGCG improves intestinal barrier function [36]. Our study showed that EGCG attenuated the impairment of intestinal tissue structural integrity and barrier function induced by *C. difficile*, as evidenced by an increase in colonic mucus and enhanced mucosal junctions in the colon. EGCG also improved the proinflammatory serum markers that induced intestinal epithelial tight junction damage. EGCG attenuated colon inflammation by decreasing serum inflammatory cytokines such as IL-6 and TNF-α in CDI mice. Increased intestinal permeability due to *C. difficile* infection leads to robust translocation of the bacterial components, resulting in endotoxemia. Our results showed increased serum LBP levels in the CDI group, while oral EGCG decreased the levels of LBP. We hypothesize that these different experimental results are linked with the intestinal microbiota and metabolism.

In addition to the protective effect on the intestinal barrier, microbes induced by EGCG promote intestinal microbiota composition and intestinal integrity [37]. One of the key ways prebiotics affect gut health is by modulating intestinal flora–immune interactions. EGCG modulates energy expenditure in CDI mice by regulating the intestinal microecosystem. Our study showed that the intestinal microbiota was regulated by EGCG and that EGCG ameliorated the decrease in community richness caused by *C. difficile* infection. Through polyphenols, beneficial bacteria are promoted in the intestine, and pathogenic bacteria are indirectly reduced. We found that large amounts of endotoxin-producing bacteria *Enterococcaceae* and *Enterobacteriaceae* were reduced by EGCG supplementation, which were linked with intestinal inflammation. Additionally, the relative abundance of these conditionally pathogenic bacteria increased in *C. difficile* patients [38]. After treatment with EGCG, the intestinal microbiota was enriched with SCFA-producing *Lactobacillaceae* and *Lachnospiraceae* in EL and EH group with *C. difficile* infection. The limitation of the study is that no samples were collected before infection, which would rasie concerns about the similarity of the microbiota between the groups. However, we conducted randomization and control variables to reduce differences. Subsequent experiments should collect a series of consecutive samples.

After EGCG intervention, alterations in the intestinal microbiota and its metabolites may be associated with improving *C. difficile* infection. EGCG improves the intestinal microbiota and its metabolites. As dietary proanthocyanidins bypass the intestine and reach the microbiota, they are biotransformed into metabolites that affect the body in a variety of ways [39]. As a result of the interaction between the gut microbiota and green tea polyphenols, SCFAs and the microorganisms that produce them may play a beneficial role in treating or preventing disease [7,40]. Additionally, SCFAs can be continuously produced by SCFA-producing bacteria if they are supplied with sufficient levels of carbon or nitrogen. The metabolic results showed that EGCG ameliorated the disruption of metabolites associated with *C. difficile* infection. Arginine and aspartic acid are strong commensal agents associated with *C. difficile* spores [41]. SCFAs, including acetate, propionate, and butyrate, are mediators of the intestinal bacteria and immune function [42]. SCFAs in feces and SCFA-producing bacteria in the gut microbiome have many health benefits, such as modulating T cells and neutrophils to reduce proinflammatory mediators [43,44]. In contrast, our experiments showed that EGCG increased the levels of SCFAs in the cecal contents of mice. Together with other beneficial effects of EGCG, some phenolic metabolites produced by it may contribute to reducing intestinal inflammatory diseases [45]. It is undeniable that mice that are orally administered EGCG benefit from the gut microbiota, functional SCFAs, and other phenolic metabolites.

Although it has been previously shown that EGCG functions as a direct antioxidant to regulate intestinal health, our experimental results suggest that this is not the only mechanism through which EGCG ameliorates *C. difficile* infection. Transcriptomic analysis showed that the protective effect of EGCG may be associated with cytokine–cytokine receptor interactions, Th17-cell differentiation, and the PPAR signaling pathway. PPARs are a group of nuclear receptors that primarily regulate various metabolic processes (e.g., fatty acid and BA metabolism) and have a variety of physiological activities such as inflammation, cell proliferation, reproduction, and cellular specialization [46]. The Th17 signaling pathway is a signaling cascade that is involved in both development and innate immunity [47,48].

## 5. Conclusions

In conclusion, we investigated the benefits of the intestinal microbiota and its metabolites regarding the effect of the prebiotic EGCG on *C. difficile* infection using a mouse model. Potential mechanisms for the effect of EGCG include the amelioration of *C. difficile*-induced alterations in inflammation pathways and the homeostasis of the intestinal microbiota and metabolome. In this study, we were able to learn more about the benefits of EGCG in intestinal inflammatory disease progression, and the bacterial taxa and genetic molecules identified may serve as potential disease biomarkers or even drug targets for future validation.

## Figures and Tables

**Figure 1 nutrients-14-03756-f001:**
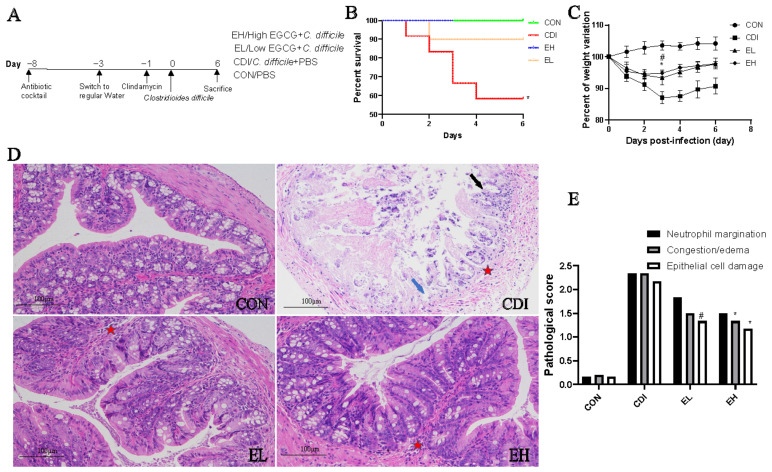
EGCG treatment improves *C. difficile*-associated colitis and increases survival. (**A**) Experimental protocol. CON group, negative control; CDI group, *C. difficile* infection; EL group, 25 mg/kg daily by gavage; EH group, 50 mg/kg daily by gavage. (**B**) Survival curves of mice in the four groups. (**C**) Body weight change curves. (**D**,**E**) Representative images and histological scores of colonic H&E staining. Blue arrow represents epithelial structure damage; black arrow represents crypt damage; asterisk represents edema and inflammation. * represents the CDI group vs. EH group, # represents CDI group vs. EL group. * and #, *p* < 0.05.

**Figure 2 nutrients-14-03756-f002:**
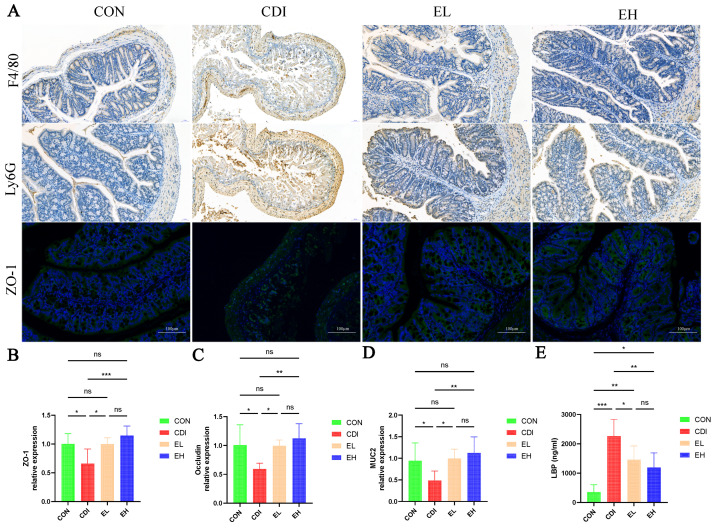
EGCG ameliorates *C. difficile*-induced impairment of the intestinal barrier. (**A**) Representative images of F4/80 and Ly6G immunohistochemical staining, and immunofluorescence staining of ZO-1 (green) and the cell nucleus (blue). (**B**–**D**) Intestinal mRNA expression (ZO-1, Occludin, and MUC2). (**E**) Serum LBP levels. *, *p* < 0.05; **, *p* < 0.01; ***, *p* < 0.001; ns, *p* > 0.05.

**Figure 3 nutrients-14-03756-f003:**
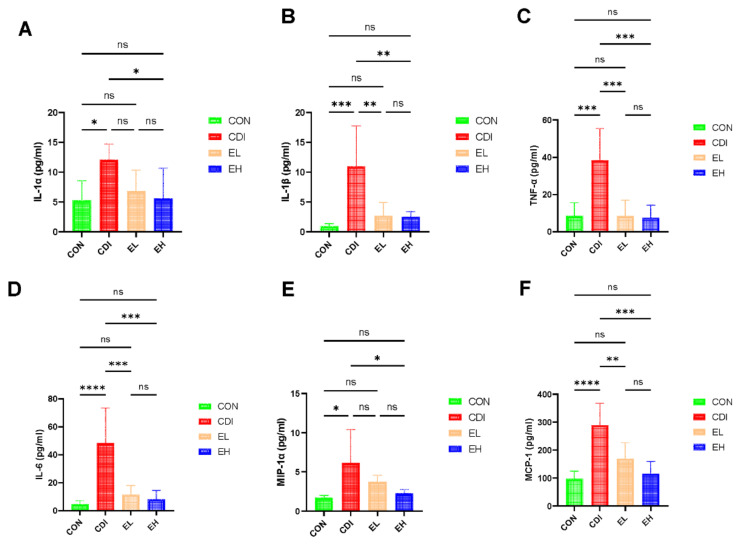
EGCG alleviates *C. difficile*-induced systemic inflammation. (**A**) IL-1α, (**B**) IL-1β, (**C**) TNF-α, (**D**) IL-6, (**E**) MIP-1α, and (**F**) MCP-1. The bars indicate the serum levels of the cytokines and chemokines in mice. *, *p* < 0.05; **, *p* < 0.01; ***, *p* < 0.001; ****, *p* < 0.0001; ns, *p* > 0.05.

**Figure 4 nutrients-14-03756-f004:**
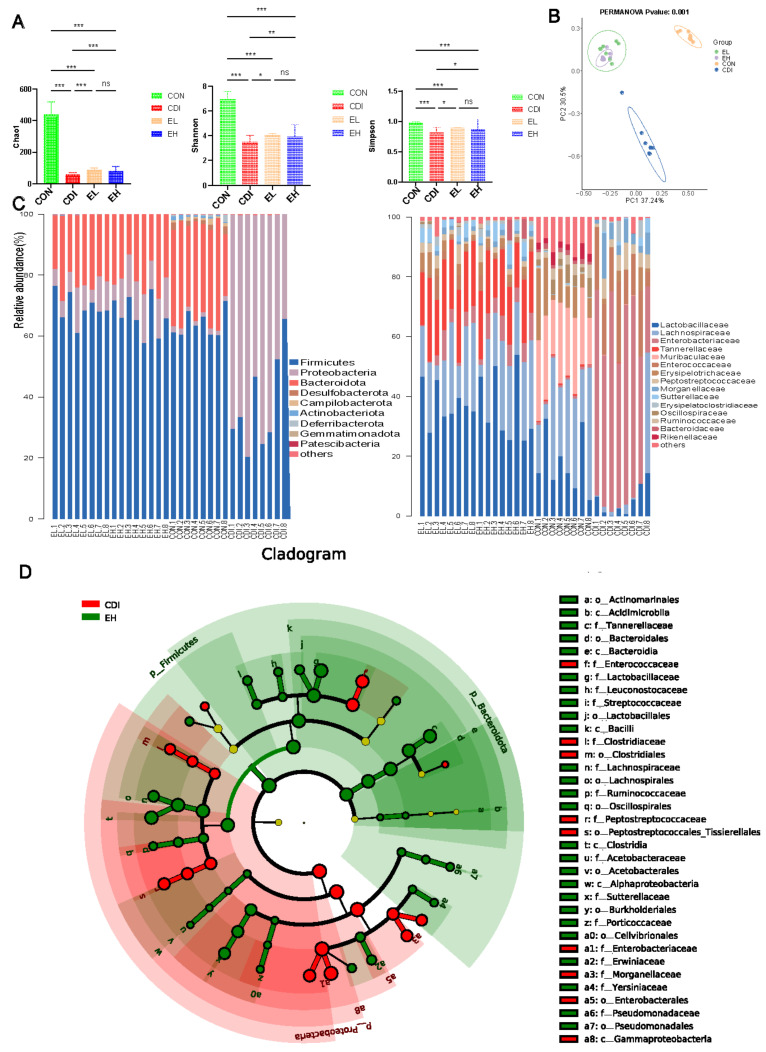
Modulation of the intestinal flora in *C. difficile*-infected mice by EGCG. (**A**) α-Diversity. (**B**) β-Diversity. (**C**) Relative levels of the most abundant taxa at the phylum and family levels. (**D**) LEfSe analysis of the EH and CDI groups. *, *p* < 0.05; **, *p* < 0.01; ***, *p* < 0.001; ns, *p* > 0.05.

**Figure 5 nutrients-14-03756-f005:**
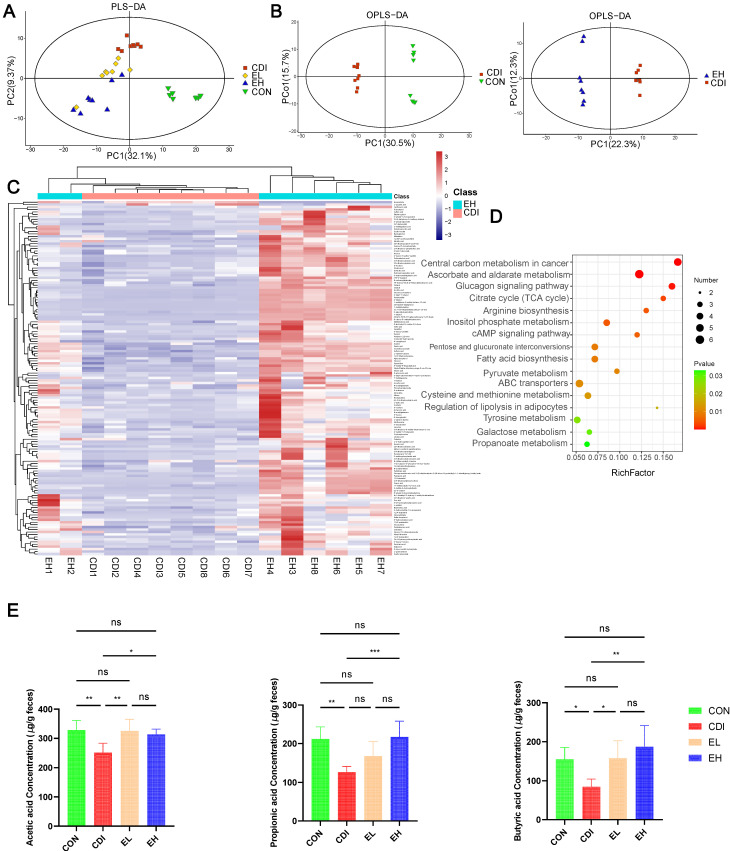
Modulation of metabolic disorders by EGCG after *C. difficile* infection. (**A**) PLS-DA showing the metabolic compositions of the CON, CDI, EL, and EH groups. (**B**) OPLS-DA plot showing the CON and CDI, CDI, and EH groups. (**C**) A hierarchical clustering heatmap of normalized (Z-score) levels visualizing significantly altered metabolites between the CDI and EH groups. (**D**) Differential KEGG metabolic pathways based on altered metabolites between the CDI and EH groups. (**E**) The levels of SCFAs between different groups. *, *p* < 0.05; **, *p* < 0.01; ***, *p* < 0.001; ns, *p* > 0.05.

**Figure 6 nutrients-14-03756-f006:**
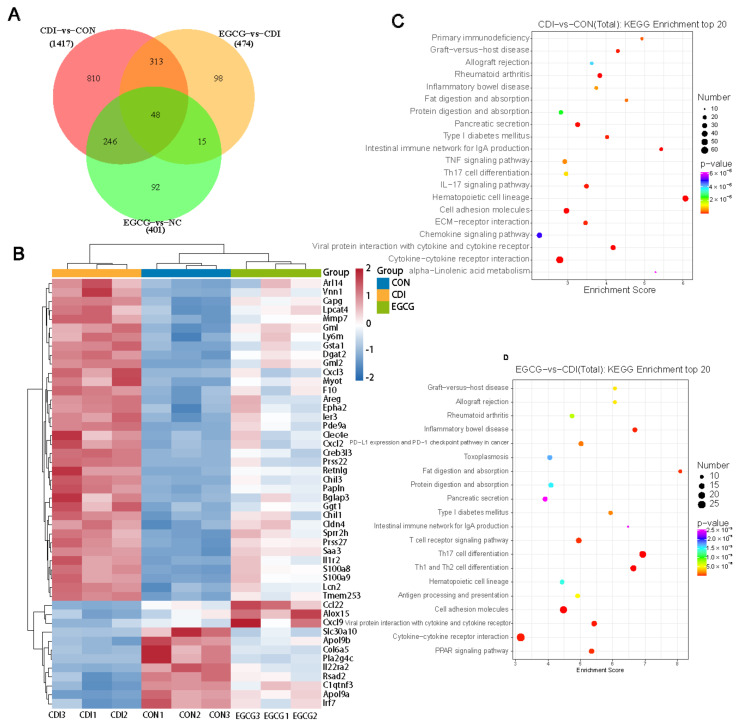
EGCG affects colonic transcriptome alterations. (**A**) Venn diagram showing the differential expression of genes between the CON, CDI, and EGCG groups. (**B**) Heatmap showing the differential expression of genes between the groups. (**C**) KEGG enrichment analysis of the CDI and NC groups. (**D**) KEGG enrichment analysis of the CDI and EGCG groups.

## Data Availability

Not applicable.

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
