# Peer review of "Epigallocatechin-3-Gallate Improves Intestinal Gut Microbiota Homeostasis and Ameliorates Clostridioides difficile Infection"

_nutrients, 2022, doi:10.3390/nu14183756_

Round 1

Reviewer 1 Report

The manuscript from Zhengjie Wu et al., studied the effects of EGCG on CDI from multiple aspects including gut microbiota. Though this is a multi-omics study, only single time point was studied, some important time points were missed. In addition, it is not well written. The introduction is not clear enough and the method is not complete enough. My detail comments are listed below:

Major:

1.     Overall, the introduction is not well written. The logic between the sentences is not clear. The effect of EGCG on the gut microbiota was not mentioned.

2.     Section 2.1 is confusing. The study design was not descripted clearly. How many mice per group? Oral administration of EGCG happed before or after CDI? When were feces collected and sequenced?

3.     There were no samples collected right before infection. Thus, it is hard to tell if the microbiota between the groups were similar before infection, which can influence the interpretation of the effect of EGCG.

4.     There were no statical tests to support the results of Figure 1b, 1c and 1e

5.     Overall, the quality of the figures is poor as font size is small and seems abnormal, and resolution is low.

6.     The basic unit of the gut microbiota is strain. Different strains in the same OTU and taxonomy may have different function and association with host. Currently, amplicon sequence variant (ASV) level analysis provides highest resolution (one base difference) in the 16S rRNA gene-based microbiota studies. Compared with OTUs, ASVs provide better resolution and more accurate results. Besides, currently most researchers generate ASVs that consider sequencing error profiles to improve specificity and sensitivity for taxonomic identification. ASV analysis should be applied in the analysis instead of OTU and taxonomic analysis. The resolution of current family level is too low.

7.     The information of sequencing data is not clear. For instance, how many reads were obtained, the depth per sample?  The processing of the sequencing data is not clear enough. For instance, the parameters of each step.

Minor:

1.     Line 19, the effect of EGCG on the gut microbiota shall be included in the introduction part in the abstract before “however”.

2.     Line 20-21, the sentence does not make sense.

3.     For people who are not the expert of Immunohistochemistry and immunofluorescence, it is hard to understand Fig 1D without more details.

4.     Line 168-169. Why cite refence 23 here?

5.     Keep the color of the groups in the figures consistent across the manuscript.

6.     Line 180-181, citation of LPS

7.     Figure 5E, code error?

8.     Line 232, “increase the relative abundance of beneficial intestinal flora” is not clear here, especially for reads who are not familiar with the bacterial names in the previous sentences.

Author Response

First, we would like to express our sincere gratitude to the reviewers for their constructive and positive comments.

Replies to Reviewer 1

The manuscript from Zhengjie Wu et al., studied the effects of EGCG on CDI from multiple aspects including gut microbiota. Though this is a multi-omics study, only single time point was studied, some important time points were missed. In addition, it is not well written. The introduction is not clear enough and the method is not complete enough. My detail comments are listed below:

Major:

  1. Overall, the introduction is not well written. The logic between the sentences is not clear. The effect of EGCG on the gut microbiota was not mentioned.

Authors’ response: Thank you for the constructive suggestions to improve our manuscript. We rewrote the section “Introduction”, please refer to our new manuscript at Line 34-60.

  1. Section 2.1 is confusing. The study design was not descripted clearly. How many mice per group? Oral administration of EGCG happed before or after CDI? When were feces collected and sequenced?

Authors’ response: Thank you for the constructive suggestions to improve our manuscript. Oral administration of EGCG once daily by oral gavage from day -8 to day 5. Fecal samples were collected from mice prior to execution. We have rewritten the Section 2.1, please refer to our new manuscript at Line 62-74.

  1. There were no samples collected right before infection. Thus, it is hard to tell if the microbiota between the groups were similar before infection, which can influence the interpretation of the effect of EGCG.

Authors’ response: Thank you for your insightful suggestion. Before the experiment, mice were allowed one weeks to acclimatize, then randomly divided into four groups based on the initial weight. We implemented these measures to ensure similar intestinal flora between groups before the experiment and to reduce differences. As to CDI group vs EH/EL group, except for C. difficile infection, EGCG was given orally once daily in the EL and EH groups, and PBS was administered once daily in the CDI group to reduce the impact of gavage operations on intestinal flora.

  1. There were no statical tests to support the results of Figure 1b, 1c and 1e

Authors’ response: We thank the reviewer for this thoughtful analysis. We have added the corresponding statical tests, please refer to our Figure 1 at Line 82.

 .

  1. Overall, the quality of the figures is poor as font size is small and seems abnormal, and resolution is low.

Authors’ response: We thank the reviewer for helpful suggestions. We have modified the pictures.

  1. The basic unit of the gut microbiota is strain. Different strains in the same OTU and taxonomy may have different function and association with host. Currently, amplicon sequence variant (ASV) level analysis provides highest resolution (one base difference) in the 16S rRNA gene-based microbiota studies. Compared with OTUs, ASVs provide better resolution and more accurate results. Besides, currently most researchers generate ASVs that consider sequencing error profiles to improve specificity and sensitivity for taxonomic identification. ASV analysis should be applied in the analysis instead of OTU and taxonomic analysis. The resolution of current family level is too low.

Authors’ response: We thank the reviewer for this thoughtful analysis. We did use ASV analysis, the method is written wrong, now has been corrected. The ASV tables has been attached to the reply letter. We apologize for our mistakes about methods. The detailed information about 16S rRNA sequencing has been added to the modified Section 2.5 and supplementary methods in Supplementary Material.

  1. The information of sequencing data is not clear. For instance, how many reads were obtained, the depth per sample? The processing of the sequencing data is not clear enough. For instance, the parameters of each step.

Authors’ response: We thank the reviewer for the helpful suggestions. We have added the information of sequencing data to the Manuscript and Supplementary material, please refer to the modified Section 2.5 at Line 108-116 and Section 3.4 at Line 210-240 and supplementary methods in Supplementary Material.

Minor:

  1. Line 19, the effect of EGCG on the gut microbiota shall be included in the introduction part in the abstract before “however”.

Authors’ response: We thank the reviewer for the helpful suggestions. We have modified the abstract, please refer to Line 19-20.

  1. Line 20-21, the sentence does not make sense.

Authors’ response: We thank the reviewer for the helpful suggestions. We have modified the Abstract.

  1. For people who are not the expert of Immunohistochemistry and immunofluorescence, it is hard to understand Fig 1D without more details.

Authors’ response: We thank the reviewer for the helpful suggestions. We have modified the Figure 1D and its figure legend.

  1. Line 168-169. Why cite refence 23 here?

Authors’ response: We cite the literature to illustrate the importance of mucins, thus showing that EGCG can protect the intestinal barrier by improving the mucin gene (MUC2).

  1. Keep the color of the groups in the figures consistent across the manuscript.

Authors’ response: We thank the reviewer for the helpful suggestions. We have modified the figures.

  1. Line 180-181, citation of LPS

Authors’ response: We have added the reference.

  1. Figure 5E, code error?

Authors’ response: We have modified the figure.

  1. Line 232, “increase the relative abundance of beneficial intestinal flora” is not clear here, especially for reads who are not familiar with the bacterial names in the previous sentences.

Authors’ response: We have modified the manuscript, please refer to Line 240

Reviewer 2 Report

In this manuscript, the authors intended to demonstrate the role of the intestinal microbiota and its metabolites in the effect of the probiotic EGCG on Clostridioides difficile infection in a mouse model. Potential mechanisms were proposed for the effects of EGCG including amelioration of C. difficile-induced alterations in inflammation-related pathways and the homeostasis of the intestinal microbiota and metabolome. Some suggestion is listed below for the consideration of revision.

1.          Introduction was too brief. The authors are suggested to enrich the related information for this study. Writing should be substantially improved.

2.          In Results, supplementation with EGCG could improve inflammation and barrier function due to CDI. As described in Introduction, 50 and 80% of the total catechin content of green tea comes from epigallocatechin-3-gallate (EGCG), and daily tea consumption could maintain microorganisms in the gut. Did the authors examine that supplementation with green tea could also improve CDI in the mouse model?

3.          Please label the days from different treatments in Fig. 1A, and add the full name of the CD in the figure legend.

4.          Please explain the definition of the pathological score in Fig. 1E and add it to the Material and Method.

5.          As shown in Fig. 2A, the authors examined the intestinal barrier protein ZO-1 using immunofluorescence analysis. Please explain the color representation in Fig. 2A, and add it to the figure legend.

6.          Please revised the labels which represent the statistical value in Fig. 5E

7.          Lines 150-153, mentioned, “…mice in the CDI group exhibited various inflammatory features, such as a thickened colonic epithelium, inflammatory cell accumulation, structural incompleteness, ruptured crypt glands, and the infiltration of inflammatory cells after infection”. Please label or quantified these results in Fig. 1 or supplementary

Author Response

First, we would like to express our sincere gratitude to the reviewers for their constructive and positive comments.

Replies to Reviewer 2

In this manuscript, the authors intended to demonstrate the role of the intestinal microbiota and its metabolites in the effect of the probiotic EGCG on Clostridioides difficile infection in a mouse model. Potential mechanisms were proposed for the effects of EGCG including amelioration of C. difficile-induced alterations in inflammation-related pathways and the homeostasis of the intestinal microbiota and metabolome. Some suggestion is listed below for the consideration of revision.

  1. Introduction was too brief. The authors are suggested to enrich the related information for this study. Writing should be substantially improved.

Authors’ response: We thank the reviewer for the helpful suggestions. We have modified the section “Introduction”, please refer to our new manuscript at Line 34-60.

  1. In Results, supplementation with EGCG could improve inflammation and barrier function due to CDI. As described in Introduction, 50 and 80% of the total catechin content of green tea comes from epigallocatechin-3-gallate (EGCG), and daily tea consumption could maintain microorganisms in the gut. Did the authors examine that supplementation with green tea could also improve CDI in the mouse model?

Authors’ response: We thank the reviewer for the helpful suggestions. Our study aims to assess the benefits of EGCG, which is a major bioactive polyphenol in green tea. There are many varieties of green tea, and we use EGCG to mitigate the differences in efficacy of the varieties. A study showed the amounts of EGCG contained in the oolong tea were 8.35 ± 0.32 μg/mL[1]. Mice consumed about 7 to 7.5 mL of water daily, which induced the amounts of active substances ingested from green tea directly is lower. As showed in our study, the high dose of EGCG make more benefits than the low dose in our CDI mouse model. Later we will conduct experiments to evaluate the benefits of green tea in CDI mouse model.

  1. Please label the days from different treatments in Fig. 1A, and add the full name of the CD in the figure legend.

Authors’ response: We thank the reviewer for the helpful suggestions. We have modified the Figure 1.

  1. Please explain the definition of the pathological score in Fig. 1E and add it to the Material and Method.

Authors’ response: We thank the reviewer for the helpful suggestions. We have added the definition of the pathological score to the Section 2.2, please refer to our new manuscript at Line 95-97.

  1. As shown in Fig. 2A, the authors examined the intestinal barrier protein ZO-1 using immunofluorescence analysis. Please explain the color representation in Fig. 2A, and add it to the figure legend.

Authors’ response: We thank the reviewer for the helpful suggestions. We have added the color representation of ZO-1 Fig. 2A to the figure legend.

  1. Please revised the labels which represent the statistical value in Fig. 5E

Authors’ response: We thank the reviewer for the helpful suggestions. We have revised the Figure 5E.

  1. Lines 150-153, mentioned, “…mice in the CDI group exhibited various inflammatory features, such as a thickened colonic epithelium, inflammatory cell accumulation, structural incompleteness, ruptured crypt glands, and the infiltration of inflammatory cells after infection”. Please label or quantified these results in Fig. 1 or supplementary

Authors’ response: We thank the reviewer for the helpful suggestions. We have revised the Figure 1E and the figure legend.

References

  1. Li, A.; Wang, J.; Kou, R.; Chen, M.; Zhang, B.; Zhang, Y.; Liu, J.; Xing, X.; Peng, B.; Wang, S. Polyphenol-rich oolong tea alleviates obesity and modulates gut microbiota in high-fat diet-fed mice. Front Nutr 2022, 9, 937279, doi:10.3389/fnut.2022.937279.

Round 2

Reviewer 1 Report

My comments on the revised manuscript as below:

Major:

1.     Though the authors used ASV in the analysis, they identified differences between the groups at genus or family level. As I mentioned before, different strains in the same taxonomy may have different function and association with host. An ASV level different abundance analysis should be performed.

2.     There were no samples collected right before infection. Though the authors said they did much to try to ensure similar intestinal flora between groups before experiment and before CDI, there was no evidence to support. At least, the authors should include this as a limitation in the discussion section.  

3.     The resolution of some figures is till poor.

4.     Regarding metabolomics, it is not clear how the authors assigned metabolites to pathways. The transcriptome analysis has the same problem.

Minor:

1.     Line 28, remove “metabolic disorder” as the following words are only related to microbial community.

2.     Line 39-41. There is a grammatical mistake in this sentence.

3.     Line 45-46. This sentence is awkward to be the last sentence in this paragraph.

4.     Line 52. This sentence introduced catechin content. However, the following sentences did not mention catechin at all.

5.     Line 55-59. Rephase these sentences. It is hard to read.

6.     Line 71-72. The conclusion is not precise.

7.     Keep the y axis title in Fig1B and 1C consistent.

8.     Line 149-150. It is not clear regarding “a Hisat2 map”, what’s the reference genome and its version?

9.     Line 154. It seems redundant after line 151-152

10.  Figure 1B.  Is there any significance between CDI and EL?

11.  Line 182. “EGCG alleviated histological changes” is not precise.

12.  Line 243-244. Got and was.

13.  Line 246. Should it be rarefication curve?

14.  Line 252. How about PERMANOVA test result?

15.  Line 259-266.  It is awkward to say xx family, or xx genus were clustered in the CDI group. What does cluster mean here?  Show the Lefse LDA score bar plot as well.

16.  Line 287. Full name of VIP

17.  The row name of figure 5C is unreadable. What were the distance and clustering method used in figure 5C?  

18.  Line 352. The first sentence is awkward. Is there any evidence related to healthy gut microbiota here?

19.  Line 423. Prebiotic?

Line 426-427. It is hard to understand.

Author Response

First, we would like to express our sincere gratitude to the reviewers for their constructive and positive comments.

Replies to Reviewer 1

Major:

  1. Though the authors used ASV in the analysis, they identified differences between the groups at genus or family level. As I mentioned before, different strains in the same taxonomy may have different function and association with host. An ASV level different abundance analysis should be performed.

Authors’ response: We thank the reviewer for the helpful suggestions. We have modified the manuscript. The statistical analysis was performed at level of phylum, family, genus, and ASV. And the top ten abundance of differential taxa were selected for relative abundance boxplot to obtain the abundance of dominant differential taxa between groups. Please refer to Line 231-232 and Figure S3 in Supplementary Material.

  1. There were no samples collected right before infection. Though the authors said they did much to try to ensure similar intestinal flora between groups before experiment and before CDI, there was no evidence to support. At least, the authors should include this as a limitation in the discussion section.  

Authors’ response: Thank you for your constructive suggestion. We have modified the manuscript, please refer to Line 353-356.

  1. The resolution of some figures is till poor.

Authors’ response: Thank you for the suggestion and we try our best to improve it.

  1. Regarding metabolomics, it is not clear how the authors assigned metabolites to pathways. The transcriptome analysis has the same problem.

Authors’ response: Thank you for your precious comment. We have added the relevant information to the manuscript, please refer to Line 268 and Line Line 303-304.

Minor:

  1. Line 28, remove “metabolic disorder” as the following words are only related to microbial community.

Authors’ response: We thank the reviewer for the helpful suggestions. We have modified the manuscript, please refer to Line 27.

  1. Line 39-41. There is a grammatical mistake in this sentence.

Authors’ response: We thank the reviewer for the helpful suggestions. We have modified the manuscript at Line 36-37.

  1. Line 45-46. This sentence is awkward to be the last sentence in this paragraph.

Authors’ response: Thank you for your precious comment. We have modified the manuscript, please refer to Line 50.

  1. Line 52. This sentence introduced catechin content. However, the following sentences did not mention catechin at all.

Authors’ response: We thank the reviewer for the helpful suggestions. We have modified the manuscript, please refer to Line 43-45.

  1. Line 55-59. Rephase these sentences. It is hard to read.

Authors’ response: Thank you for your precious comment. We have modified the manuscript, please refer to Line 47-54.

  1. Line 71-72. The conclusion is not precise.

Authors’ response: We thank the reviewer for the helpful suggestions. We have modified the manuscript, please refer to Line 62-63.

  1. Keep the y axis title in Fig1B and 1C consistent.

Authors’ response: We thank the reviewer for the helpful suggestions. We have modified the Fig. 1.

  1. Line 149-150. It is not clear regarding “a Hisat2 map”, what’s the reference genome and its version?

Authors’ response: We thank the reviewer for the helpful suggestions. We have modified the manuscript, please refer to Line 137-138.

  1. Line 154. It seems redundant after line 151-152

Authors’ response: We thank the reviewer for the helpful suggestions. We have modified the manuscript, please refer to Line 140-141.

  1. Figure 1B.  Is there any significance between CDI and EL?

Authors’ response: We thank the reviewer for the helpful suggestions. There is no difference between CDI and EL group. We have modified the manuscript, please refer to Line 154-155.

  1. Line 182. “EGCG alleviated histological changes” is not precise.

Authors’ response: We thank the reviewer for the helpful suggestions. We have modified the manuscript, please refer to Line 165.

  1. Line 243-244. Got and was.

Authors’ response: Thank you for the comment. We have modified the manuscript, please refer to Line 220-221.

  1. Line 246. Should it be rarefication curve?

Authors’ response: We thank the reviewer for the helpful suggestions. We have modified the manuscript, please refer to Line 223.

  1. Line 252. How about PERMANOVA test result?

Authors’ response: We thank the reviewer for the helpful suggestions. We have added PERMANOVA test result to the manuscript, please refer to Line 229.

  1. Line 259-266.  It is awkward to say xx family, or xx genus were clustered in the CDI group. What does cluster mean here?  Show the Lefse LDA score bar plot as well.

Authors’ response: We sincerely appreciate the valuable comments. We have modified the manuscript, please refer to Line 237, Line 241, Line 243, and Figure S5 in Supplementary Material.

  1. Line 287. Full name of VIP

Authors’ response: We thank the reviewer for the helpful suggestions. We have modified the manuscript, please refer to Line 261-262.

  1. The row name of figure 5C is unreadable. What were the distance and clustering method used in figure 5C?  

Authors’ response: We thank the reviewer for the helpful suggestions. We have modified the manuscript, please refer to Line 281-282 and Figure S7 in Supplementary Material.

  1. Line 352. The first sentence is awkward. Is there any evidence related to healthy gut microbiota here?

Authors’ response: Thank you for the comment. We have added the reference to the manuscript, please refer to Line 318.

  1. Line 423. Prebiotic?

Line 426-427. It is hard to understand.

Authors’ response: We sincerely appreciate the valuable comments. We have modified the manuscript, please refer to Line 389 and Line 393.

Reviewer 2 Report

The authors did a good job in answering all the questions I raised in the original version. The manuscript has been substantially improved, and thus recommended for acceptance.  

Author Response

Thank you for the help.